# Experimental Studies on the Effect of Properties and Micro-Structure on the Creep of Concrete-Filled Steel Tubes

**DOI:** 10.3390/ma12071046

**Published:** 2019-03-29

**Authors:** Rongling Zhang, Lina Ma, Qicai Wang, Jia Li, Yu Wang, Huisu Chen, Valeriia Samosvat

**Affiliations:** 1School of Civil Engineering, Lanzhou Jiaotong University, Lanzhou 730070, China; wangqc@mail.lzjtu.cn (Q.W.); lijiageotech@hotmail.com (J.L.); samosvat.valeriia@gmail.com (V.S.); 2Cardiff School of Engineering, Cardiff University, Cardiff CF24 3AA, UK; WangY219@cardiff.ac.uk; 3Department of Civil, Construction, and Environmental Engineering, Iowa State University, Ames, IA 50011, USA; 4Jiangsu Key Laboratory of Construction Materials, School of Materials Science & Engineering, Southeast University, Nanjing 211189, China; chenhs@seu.edu.cn

**Keywords:** concrete-filled steel tubes (CFSTs), lateral restraint, creep, pore structure, mechanics

## Abstract

To study different lateral restraints, different constituents of expansion agents, the influence of different steel ratios, and concrete creep properties, we carried out experiments with lateral restraint and without lateral restraint conditions separately on 12 specimens with the expansion agent content accounting for 4%, 8%, and 12% respectively. In addition, the creep tests were performed on specimens with different steel ratios of 0.0%, 3.8%, 6.6%, and 9.2%. The test results show that the lateral restraint improves the strength of the system (concrete-filled steel tubes) which resists further load after the concrete ultimate strength is surpassed and reduces the creep. The creep degree of the concrete-filled steel tube with lateral restraint is about 0.09–0.30 times smaller than that of the tube without lateral restraints. The creep degree of the concrete-filled steel tube increases as the steel ratio decreases. Creep tests with different amounts of expansion agent indicate that the creep degree of the concrete structure increases as expansion agent content decreases. To study the internal mechanism of the creep of concrete-filled steel tubes with different lateral restraints and different expansion agent concentrations, a microscopic pore structure test on the steel core concrete was conducted using the RapidAir457 pore structure instrument. Microscopic studies show that the air content and the length of the bubble chord of the laterally restrained core concrete are lower than those without lateral restraint core concrete. The amount of air content and the length of the bubble chord of core concrete specimens increase as the expansion agent content in the core concrete specimens decreases from 12% to 4%. Under the same external loading conditions, as steel ratio increases, the lateral restraint causes a further reduction of creep. The results of this study suggest that the creep of concrete can be reduced by selecting appropriate lateral restraint conditions and an optimal amount of expansion agent in the mix design of concrete for concrete-filled steel tubes.

## 1. Introduction

A concrete-filled steel tube (CFST) is an effective composite structure that is manufactured with advanced technologies to combine and optimize both the high strength and ductility of steel and the high-pressure resistance of concrete in the composite structure. According to the triaxial stress state of concrete specimens under loading, the lateral restraints of steel tubes can improve the compressive strength and the elasticity modulus and increase the plasticity of concrete, especially for high-strength and ultra-high-strength concrete. Such concrete has been widely used in the composite structures in steel tubes, which provide lateral restraints to increase the toughness and reduce the brittleness of the steel structures. The composite structures are manufactured entirely by filling the steel tubes with concrete to avoid local buckling in the steel. A CFST is the best combination of two materials that are concrete and steel tube [1] and has significant advantages in mechanical properties compared to other structures [2,3,4]. Therefore, a CFST has been widely used in long-span bridges, high-rise buildings, multi-story buildings in industrial plants, large airports, and overhead power transmission towers. Concrete creep has an adverse effect on concrete deformation under a sustained load over a long period.

In the prestressed concrete structure, concrete creep causes a loss of prestress of the structure by redistributing internal forces that directly affect the long-term performance and shorten the service lifetime of the prestressed concrete. Long-term creep is a significant material response that needs to be accounted for in the concrete structural design [5]. Therefore, study on the destructive mechanism of concrete creep with efforts to reduce its adverse effect has become a major concern in engineering applications of concrete. Both experimental and theoretical analyses have been carried out on concrete shrinkage and creep [6,7,8,9,10,11,12,13]. Charpin et al. [14] presents a 12-year-long creep and shrinkage experimental campaign on cylindrical and prismatic concrete samples under uniaxial and biaxial stress, respectively. Wendner [15] proposes the B4 model, based on the B3 model, by taking account of cement types, admixture concentrations, and aggregate composition on concrete shrinkage and creep. This method aims to overcome disadvantages in concrete creep and provide predictions using CEB-FIB1990, ACI209, and GL2000 models. The research results from Saeed Rahimi-Aghdam et al. [16] indicated that the age effect on creep at variable humidity and temperature histories is based not on an empirical effective age, but on the effective hydration time calculated from the increase in hydration degree at each point of the structure. Anders [17] proposes a model that can predict the basic creep and relaxation of concrete. Cao et al. [18] conducted a creep test on cylindrical concrete specimens at different ages and suggest that loading age has a greater impact on the creep coefficient. Gholampour et al. [19] showed that concrete mixes containing recycled lower strength concrete aggregates show higher creep deformation compared to mixes prepared with recycled higher strength concrete aggregates. Harinadha’s research results indicate that the higher the aggregate-to-cement ratio is, the lower the creep is [20]. Müller [21] shows that the creep of hardened cement paste under load leads to an increase of the inner specific surface area of the material, which is likely related to inter-crystalline processes. Neville [22] states that the creep mechanism can be interpreted using viscoelastic theory, seepage theory, viscous flow theory, the internal force equilibrium theory, and micro fracture theory, among others. The research results from Yu et al. [23] indicated that materials with finer particles show less creep. they studied the influence of the microstructure on the macroscopic creep of concrete. Although the creep and the resulting shrinkage in concrete structures have been studied in depth, the creep mechanism of CFSTs is not well understood. Due to the increasing use of CFSTs in current civil constructions, there is a need to understand the creep properties of CFSTs under the long periods of loading. The creep effect of CFSTs has become a major concern in engineering applications, especially in structural design. Zhong [1] proposes an empirical equation for calculating the creep deformation of axial compression for short columns. Han [24] performed a theoretical analysis of the creep characteristics of a CFST’s axial compression with the age-adjusted effective modulus of elasticity method. Chen [25] conducted experiments on 22 specimens of CFST short columns to study the effects of the recycled coarse aggregate replacement rate and the ferrule index on bearing behavior and deformation performance. Some researchers [26,27,28] have studied the effects of three different expansion agents in concrete mixtures and five different stress ratios by conducting creep tests on CFST specimens.

CFSTs are composite structures. Steel tubes provide lateral restraints inside concrete structures in multiple directions. Concrete is a typical heterogeneous porous medium composed of solid materials (e.g., sand, stone, and cement), liquid (water), and gas (air). Expansion agents are often added to concrete mixtures to provide micro expansion and to limit shrinkage of the concrete, making the steel tube tightly contact concrete and avoiding any debonding between the two materials. The effects of expansion agents and the lateral restraint effect by steel tubes on creep are different from those of ordinary concrete. Lateral restraint conditions, expansion agent amounts, and steel ratios are recognized as key factors that cause creep and shrinkage in CFST structures. However, the deformation mechanism caused by these factors has not yet been explored

This paper mainly investigates the creep of CFSTs with two lateral restraint conditions (with and without lateral restraints), three expansion agent concentrations (4%, 8%, and 12%), and four steel ratios (0%, 3.8%, 6.6%, and 9.2%). Air void characteristics in concrete with or without lateral restraints, mixed with different concentrations of expansion agents, were also studied, and air-void parameters were determined with a RapidAir457 analyzer (Copenhagen, Denmark). The main goal was to explore the mechanism of creep under combination conditions of varied lateral restraint, different expansion agent concentrations, and different steel ratios. The outcomes of this study can contribute to a better understanding of the long-term mechanical performance of CFSTs in structural infrastructures.

Highlights:A 480-day experiment on the creep of concrete with lateral restraints was carried out.It was observed that the lateral restraint greatly reduces the creep degree in concrete.The influence of the mechanism of lateral restraint on creep was analyzed by means of the microstructure of concrete.

## 2. Materials and Methods

### 2.1. Materials

Mechanical properties of raw materials were measured before every test. Fine aggregates were of natural sand with a fineness modulus of 3.2, where the apparent density was 2716 kg/m^3^ and the mud content was 0.5%. The coarse aggregate was demolished basalt with particle sizes within a range of 5–20 mm, apparent density was 2680 kg/m^3^, crushing index was 3.2%, and alkali activity was 0.07. The properties of cement, fly ash, mineral powder, expansion agent, air-entraining agent, and water reducing agent are summarized in Table 1, Table 2, Table 3 and Table 4. A Q345 round steel tube (Baotou Iron and Steel (Group) Co., Ltd., Baotou, China, the yield strength was 345 MPa) was used to provide lateral restraints. Mechanical properties of the steel tube are summarized in Table 5.

### 2.2. Experimental Design

All steel tubes used in this study are 140 mm in diameter and 350 mm in length, but the thickness of the tubes varied by 1.3, 2.2, and 3.0 mm, which resulted in differences in the steel ratio of the CFST of 3.8%, 6.6%, and 9.2%, respectively. In addition, the water cement ratio was 0.3, whereas the expansion agent content in the core concrete varied by 4%, 8%, and 12%. Twelve specimens were prepared with a combination of two lateral restraint conditions, three varied expansion agent concentrations, three steel ratios, and two diameter sizes. All the information about each specimen including mix design of the concrete is summarized in Table 6. At 28 days after casting, creep, and shrinkage measurement was conducted on each specimen. The creep test equipment, which followed the GB/T 50082-2009 standard [29] is shown in Figure 1. A load of 235.3 kN was applied to the CFST specimens to simulate long-term loading conditions. The concrete structure without lateral restraints was tested by the same stress method. The stress distribution of CFST was calculated according to the combination structure. The coupling effect of the lateral restraint on the concrete was considered, and the stress of the core concrete was determined. Stress characteristics of the composed structures were considered in computing total stress on the core concrete. To ensure the same stress was applied on both concrete specimens, with lateral restraints and without, a load of 170.1 kN was imposed on the concrete without lateral restraints. The concrete specimen without lateral restraints was entirely sealed with tape pre- and post-creep test. The laterally restrained specimen was only sealed with tape at both ends of the specimen.

A dial indicator with an intelligent memory (JMZX-212A, Kingmach Measurement&Monitoring Technology Co., Ltd, Changsha, China) and a string strain gauge were mounted on the specimen to measure deformation stress, and vertical deformation at the middle of the specimen (Figure 2 and Figure 3) while maintaining a room temperature of 20 ± 1 °C. A schematic sketch is used to demonstrate the test rig (Figure 1), the laterally restrained specimen (Figure 2), and the specimen without lateral restraint (Figure 3).

## 3. Analysis of Test Results

### 3.1. Effect of Lateral Restraints on the Creep of Concrete

Effects under varied lateral loading were studied in two ways. First, the creep properties of the concrete specimens with and without lateral loading were compared. The steel tube was fixed on the concrete specimen to provide lateral restraint. Second, the lateral loading concrete with different steel ratios, which can result in different degrees of lateral restraint, was investigated. The results of all 12 creep tests are shown in the three subfigures of Figure 4. Each subfigure represents a scenario in which a given expansion agent content was applied to each group of four specimens. The creep results between two specimens under varying lateral loadings are summarized in Table 7.

According to the test results and analysis shown in Figure 4 and summarized in Table 7, the degree of creep in a CFST with lateral restraint is about 0.08–0.23 times smaller than that of concrete without lateral restraint when all concrete mixture contains 4% expansion agent; the degree of creep in a CFST with lateral restraint is about 0.09–0.29 times smaller than that of concrete without lateral restraint when all four concrete mixture contains 8% or 12% expansion agent separately. Based on comparisons of the degrees of creep between specimens with and without lateral restraint under the given loading for up to 480 days, lateral restraints effectively reduce the degree of creep in concrete and improve its long-term mechanical properties.

Moreover, creep variation caused by different steel ratios were also compared in pairs:When all specimens had 4% expansion agent, the creep ratio of the specimen with a steel ratio of 6.6% was about 0.42–0.71 times that with a steel ratio of 3.8%, while the creep ratio of the specimen with a steel ratio of 9.2% was about 0.44–0.58 times that with a steel ratio of 3.8%.When all specimens had 8% expansion agent, the creep ratio (coefficients) of the specimen with steel ratio of 6.6% was about 0.48–0.74 times that with a steel ratio of 3.8%, while the creep ratio (coefficients) of the specimen with a steel ratio of 9.2% was about 0.44–0.48 times that with a steel ratio of 3.8%.When all specimens had 12% expansion agent, the creep ratio (coefficients) of the specimen with a steel ratio of 6.6% was about 0.51–0.76 times that with a steel ratio of 3.8%, while the creep ratio (coefficients) of the specimen with a steel ratio of 9.2% was about 0.45–0.55 times that with a steel ratio of 3.8%.

In this paper, based on the ferrule effect of the combined structure, the mechanical properties of the concrete under different lateral restraints are calculated based on the confinement effect of the combined structure. The influence of the steel tube’s lateral restraints on the creep characteristics of the concrete was studied via detailed calculation and analysis considering the load distribution between concrete and steel tube. The results of the analysis are summarized in Table 8. The standard confinement coefficient and the standard value of combined compressive strength increased with increasing steel ratio. The stress ratio of the CFST specimen decreased as the steel ratio increased. The distribution coefficient of concrete load increased and that of the steel tube load decreased with increasing steel ratio. That is, the difference in lateral restraints caused the difference in stress distribution between the concrete and steel tubes. This also shows why lateral restraint influences the creep characteristics of concrete.

All creep tests on CFST specimens with different steel ratios show that, except for the first 10 days, the creep ratio (coefficients) decreased as the steel ratio increased up to 9.2%. The creep ratio (coefficients) in the first 10 days did not decline when the steel ratio increased from 6.6 to 9.2% in the concrete with 4% and 12% expansion agent. The reason for this phenomenon is that lateral restraint enhances with the increase in steel ratio. Lateral restraint improves the strength of the core concrete. Furthermore, the steel tube improves the lateral restraint by increasing the steel ratio, which effectively shares the external load. Consequently, less stress in the core concrete leads to a creep ratio (coefficients) reduction under the same external loads. Triaxial stresses were applied to the specimen because of a combination of both lateral constraints and vertical loading, and these stresses affect the strength of the core concrete, which in turn affects creep behavior.

### 3.2. The Effect of Expansion Agent on the Creep of the CFST

The results are summarized in four groups in Figure 5. Each group indicates a certain lateral restraint. Thus, the differences in creep ratio (coefficients) with varied concentrations of expansion agent are shown.

Creep ratios of the two concrete specimens with different expansion agent concentrations are compared in pairs in Table 9. The outcomes can be generally summarized as follows:When the steel ratio was 3.8%, with expansion agent concentrations of 8% and 12%, the creep ratios were about 0.865–0.910 and 0.825–0.863 times those of the 4% expansion agent, respectively, except for the first 240 h.When the steel ratio was 6.6%, with expansion agent concentrations of 8% and 12%, the creep ratios were about 0.895–0.932 and 0.869–0.964 times those of the 4% expansion agent, respectively, except for the first 2400 h.When the steel ratio was 9.2%, with expansion agent concentrations of 8% and 12%, the creep ratios were about 0.932–0.976 and 0.873–0.915 times those of the 4% expansion agent, respectively, except for the first 240 h.When the concrete specimens contained 8% and 12% expansion agent but no lateral loading, the creep ratios were about 0.841–0.886 and 0.778–0.817 times those of the 4% expansion agent, respectively.

In general, increasing expansion agent concentrations in the concrete effectively reduced the creep ratio as lateral loading was fixed. However, when the expansion agent concentration increased from 8% to 12%, the reduction of the creep ratio was much smaller than when the agent concentration increased from 4% to 8%. In other words, the decrease in creep ratio was not proportional to the increase in expansion agent concentration. Therefore, there is an optimum rate of expansion agent that brings about an optimal effect on creep ratio.

## 4. The Analysis of the Micro Pore Structure

Eighteen cube specimens (there are three samples in each group, six groups in total) with a side distance of 100 mm that have the same lateral restraint conditions and expansion agent concentrations as the creep test specimens were cured for 28 days and prepared for pore structure analyses. Figure 6 shows that the cube specimens were curing for pore structure analysis. Plastic film was used to seal all the cube specimens to simulate sealing effect of steel tubes on the core concrete of the CFST specimens. Concrete specimens at the bottom layer were not demolded to provide the lateral restraints to simulate the effect of steel tube on the core concrete. Concrete specimens at the top two layers were demolded to simulate the ordinary concrete without lateral restraints. Moreover, the top two layers provided vertical restraint for the bottom layer. Finally, the pore structures of the concrete specimens were analyzed at the age of 28 days. The procedure was as follows: In the first place, each specimen was cut into three specimens before their surface was lapped and polished. Afterwards, the specimens were placed in the oven to dry. Secondly, the surface of each specimen was coated by a layer of ink. The second layer containing Al_2_O_3_ was applied to the specimen after the ink dried. Afterwards, Al_2_O_3_ was removed, and the specimen was placed under an air-void analyzer (RapidAir457) for parameter measurement. The test equipment with specimens is shown in Figure 7. Figure 8 demonstrates a series of test results.

### 4.1. Effect of Lateral Restraints on Concrete Pore Structure under Different Conditions

Each subfigure in Figure 9 shows the curves of the bubble chord length, the air content, and the number of bubbles with the same expansion agent, and different lateral loading scenarios are compared.

As Figure 9a shows that, for concrete specimens under lateral restraint with an expansion agent concentration of 4%, the air content and bubble chord length are 0.85 and 0.82 times larger than their counterparts without lateral restraint, respectively. When the expansion agent concentration of specimens with lateral restraint increases to 8%, the air content and bubble chord length are 0.82 and 0.88 times larger than their counterparts without lateral restraint, respectively, as shown in Figure 9b. Further increase in expansion agent concentration to 12%, as shown in Figure 9c, the air content and bubble chord length of specimens with lateral restraint are 0.76 and 0.80 times larger than those of specimens without lateral restraint, respectively. Within the same range of bubble chord size, the percentage of air content and the number of bubble chords in the laterally restrained core concrete are less than those of specimens without lateral restraint. These pore structure analyses suggest that lateral restraint helps to develop the crystallization of the expansion agent, reducing air content and the number of chords in the concrete structure.

The self-stress effect of the core concrete with lateral restraint contributes to further hydration of the expansion agent in the concrete structure. Therefore, there will be more crystallized expansion agent to make the concrete structure denser with lateral restraints than without lateral restraints. This also explains why creep ratio is smaller for the concrete with lateral restraint than that without lateral restraint, as shown in Figure 5. Analyses of these results suggest that lateral restraint is the most important factor affecting concrete creep.

### 4.2. Effect of the Concentration of Expansion Agent Content on the Core concrete Pore Structure

Figure 10 shows the bubble chord size, the air content, and the number of bubble chords when the concrete specimens have three different expansion agent concentrations (4%, 8%, and 12%) with lateral restraint. Figure 10a,b show the details under different scenarios without and with lateral restraint.

Air content and the number of bubble chords with an expansion agent concentration of 8% (or 12%) are 0.63 and 0.78 times (or 0.84 and 0.75 times) higher than those with an expansion agent concentration of 4%, respectively, for concrete specimens with lateral restraint, as Figure 10a shows. Figure 10b illustrates scenarios with lateral restraint. Air content and the number of bubble chords with an expansion agent concentration of 8% (or 12%) are 0.61 and 0.84 times (or 0.64 and 0.86 times) larger than those with an expansion agent concentration of 4%, respectively. Results in Figure 10 reveal that the percentage of air content and the number of bubble chords decrease with increasing expansion agent content under the same lateral restraint. Chemical reaction of the expansion agent generates ettringite to fill in the cracks and voids in the core concrete. The crystallization pressure promotes further hydration of the concrete and, with the addition of expansion agent, makes the core concrete structure denser than that without expansion agent. These functions of expansion agents in core concrete structures also explain why creep ratios decrease as expansion agent concentration increases, as shown in Figure 6. Therefore, in addition to lateral restraint, the expansion agent ratio is an important factor affecting the creep of core concrete.

## 5. Conclusions

This paper investigates the creep behavior in CFSTs. Specimens with different expansion agent concentrations and different steel ratios were tested, and the results were compared in pairs. Meanwhile, air-void analysis in the concrete structure in a microscope is also carried out. The following conclusions are drawn from this study:Analyses on 12 creep test results reveal that creep ratios of the concrete specimens with lateral restraint are about 0.09–0.30 times smaller than those without lateral restraint. Meanwhile, the creep ratio of CFST specimens decreases as the steel ratio increases.Creep ratio decreases as expansion agent concentration increases.According to analyses on the microscopic pore structures of concrete specimens, the percentage of air content and the number of bubble chords of the core concrete with lateral restraint are lower than those without lateral restraint. Air content and the number of bubble chords reach the highest values in the core concrete with a 4% expansion agent, followed by core concrete with 8% expansion agent and then with 12% expansion agent.The lateral restraint and the expansion agent concentrations have a combined effect on the creep of the core concrete, so selecting the appropriate lateral restraint and the best content for the expansion agent can effectively reduce the creep of concrete.The key contribution of this study is the finding that CFSTs require an optimum combination of proper lateral restraint and an optimum ratio of expansion agent in order to effectively reduce the creep and thus improve the long-term service of concrete structures.

## Figures and Tables

**Figure 1 materials-12-01046-f001:**
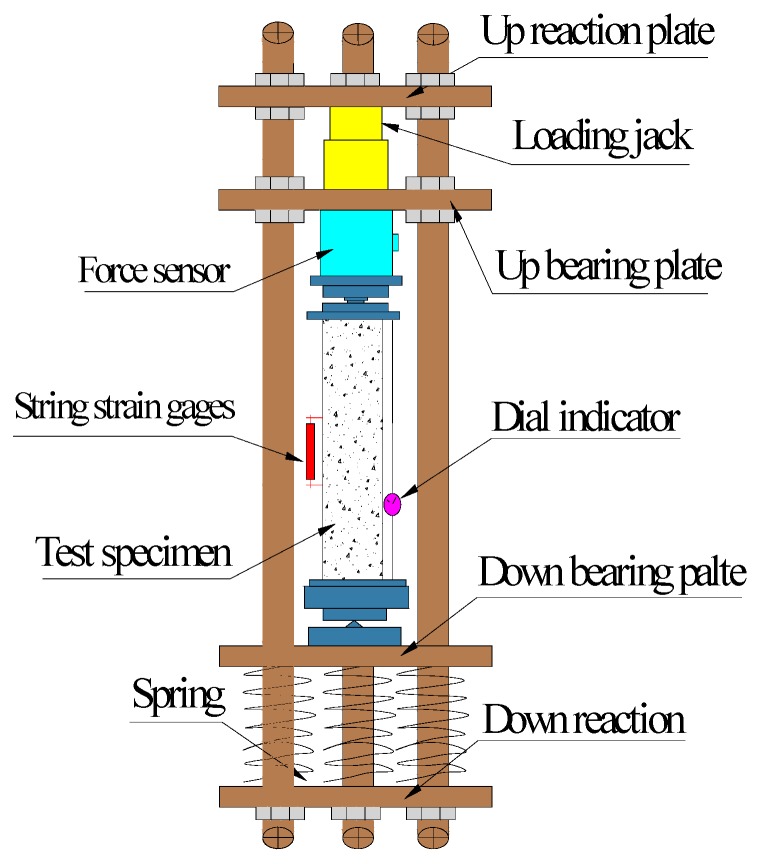
Schematic diagram of the creep test setup.

**Figure 2 materials-12-01046-f002:**
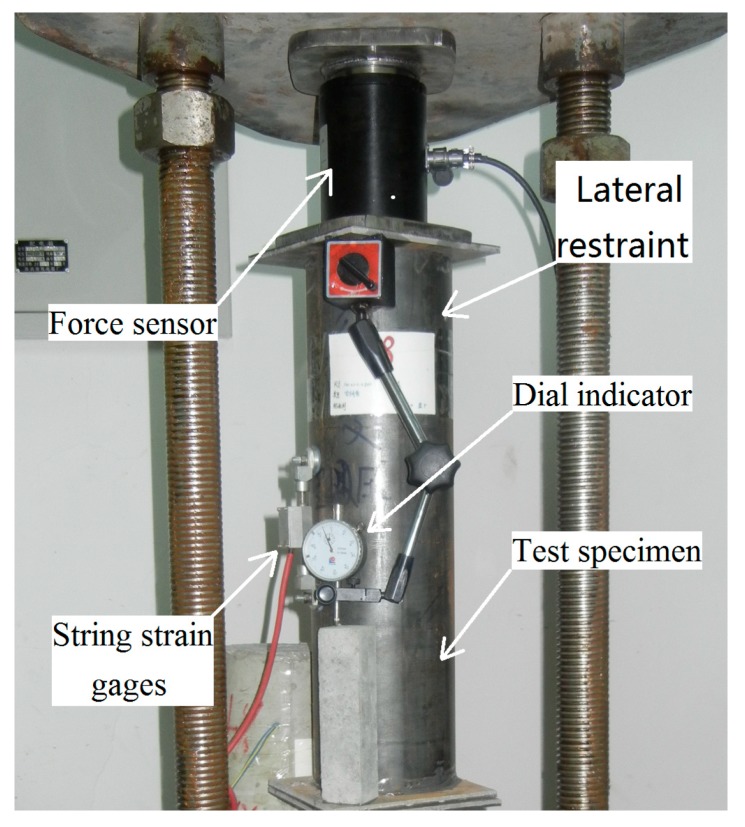
Example of a creep test on the specimen with lateral restraint.

**Figure 3 materials-12-01046-f003:**
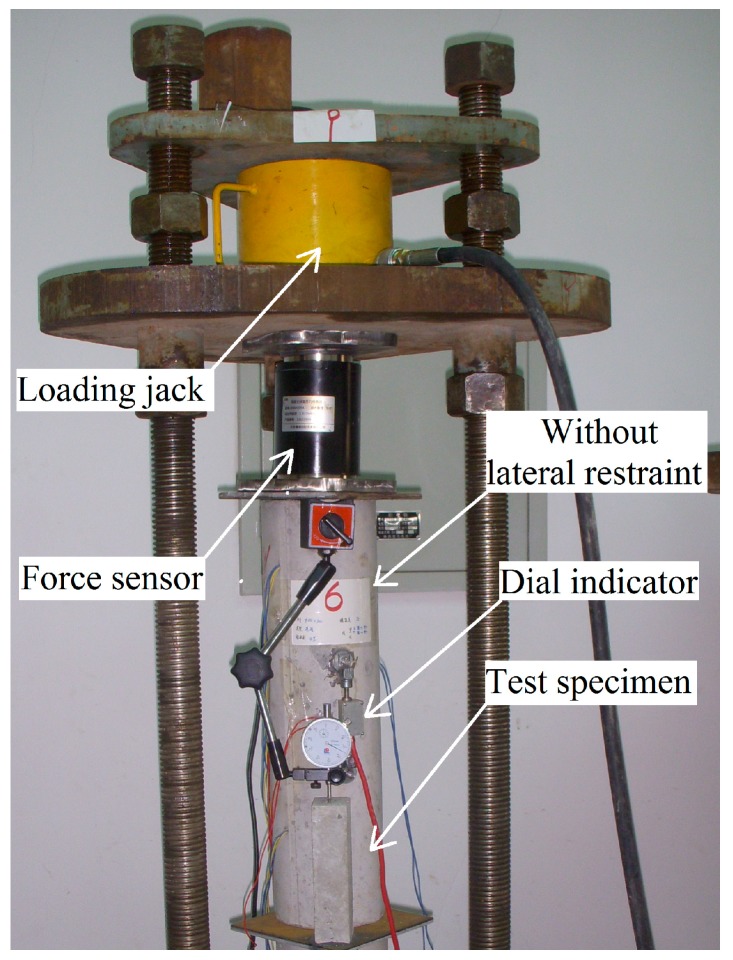
Example of a creep test on the specimen without lateral restraint.

**Figure 4 materials-12-01046-f004:**
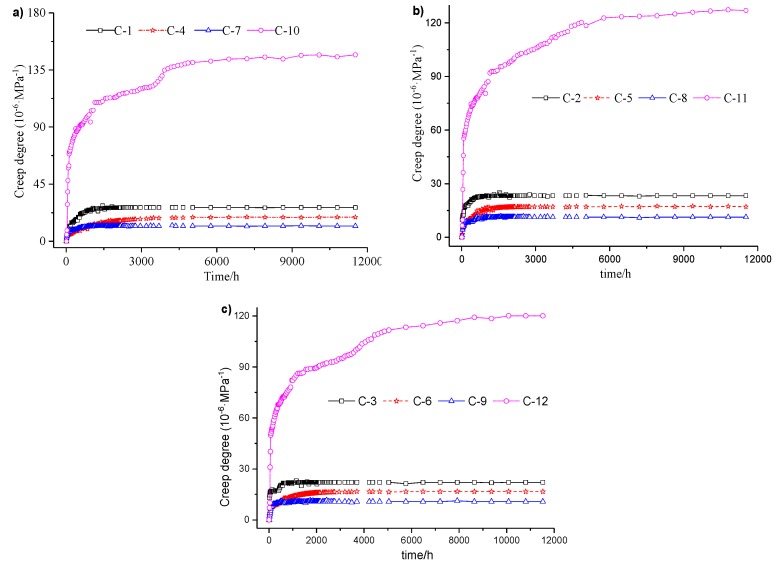
Creep of concrete with different lateral restraint conditions: (**a**) expansion agent dosage is 4%; (**b**) expansion agent dosage is 8%; (**c**) expansion agent dosage is 12%.

**Figure 5 materials-12-01046-f005:**
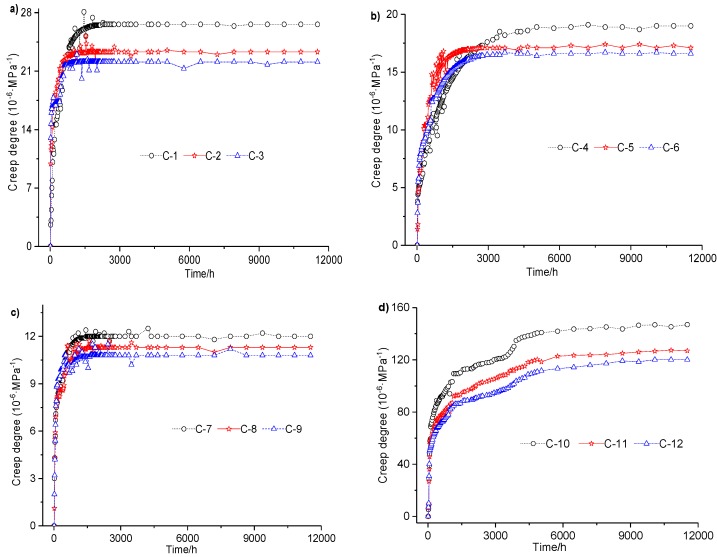
The creep of concrete with three different expanding agent concentrations: (**a**) the steel ratio is 3.8%; (**b**) the steel ratio is 6.6%; (**c**) the steel ratio is 9.2%; (**d**) the steel ratio is 0%.

**Figure 6 materials-12-01046-f006:**
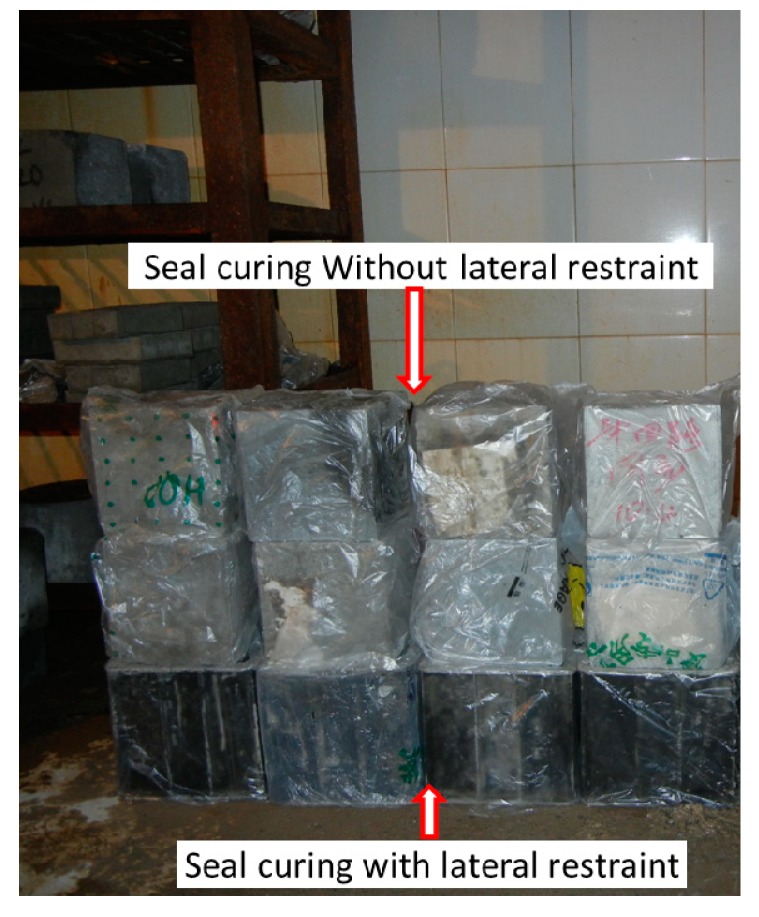
Concrete test specimens with different lateral restraints.

**Figure 7 materials-12-01046-f007:**
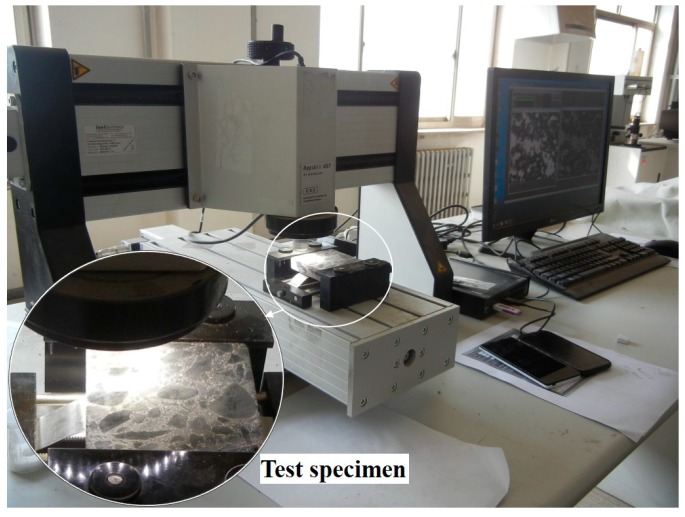
The RapidAir457 tester analyzing pore structure.

**Figure 8 materials-12-01046-f008:**
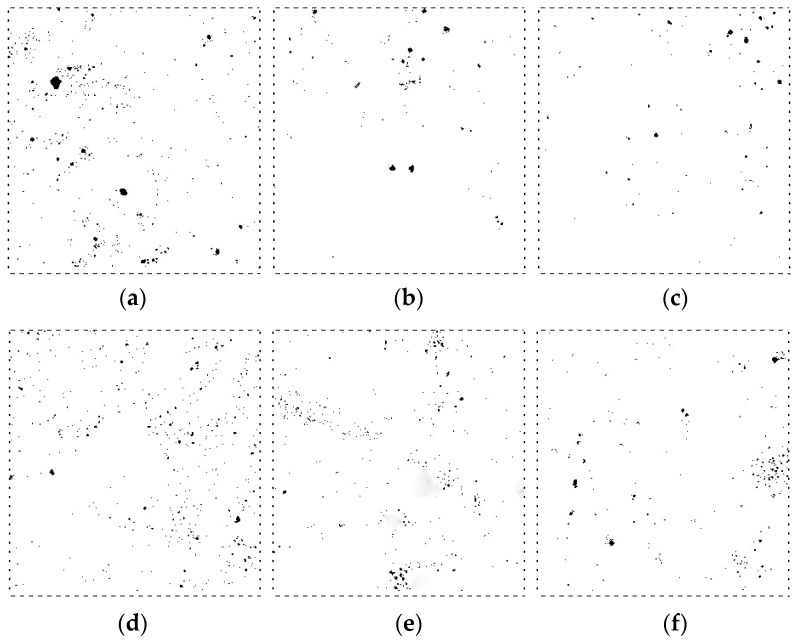
Concrete pore structure under different conditions: (**a**) expansion agent dosage is 4% and without lateral restraint; (**b**) expansion agent dosage is 8% and without lateral restraint; (**c**) expansion agent dosage is 12% and without lateral restraint; (**d**) expansion agent dosage is 4% and lateral restraint; (**e**) expansion agent dosage is 8% and lateral restraint; (**f**) expansion agent dosage is 12% and lateral restraint.

**Figure 9 materials-12-01046-f009:**
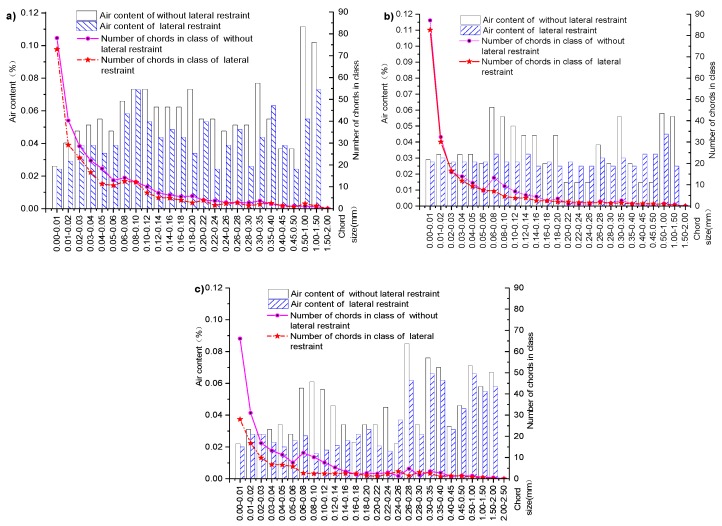
Effects of lateral restraint conditions on core concrete pore structures: (**a**) 4%; (**b**) 8%; (**c**) 12%.

**Figure 10 materials-12-01046-f010:**
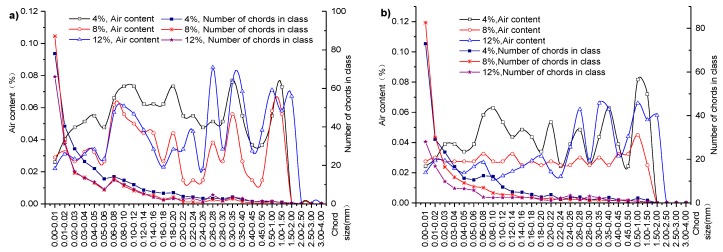
Effect of the three varied expansion agent concentrations on core concrete pore structures: (**a**) without lateral restraint; (**b**) with lateral restraint.

**Table 1 materials-12-01046-t001:** Performance index summary of silicate with low alkaline cement.

Material	Volume Stability (mm)	Sulfur Trioxide (%)	Chloride Ion (%)	Alkali Content (%)	Loss on Ignition (%)	Initial Setting (min)	Final Setting (min)	Compressive Strength (MPa)	
								3 d	28 d
Cement	2	2.44	0.012	0.43	1.52	185	325	21.7	48.6

**Table 2 materials-12-01046-t002:** Performance index summary of slag and fly ash.

Material	Specific Surface Area (m^2^/kg)	Chloride Ion Content (%)	Water Content (%)	Sulfur Trioxide (%)	Free Calcium Oxide (%)	Fluidity Rate (%)	Fineness (%)	Water Demand (%)	Alkali Content (%)	Loss on Ignition (%)	Activity Index (%)	
											H7	H28
Slag	448	0.016	–	1.90	–	97	–	–	0.36	0.19	79	97
Fly ash	-	0.01	0.25	1.4	0.6	–	8.6	82	0.81	3.2	–	–

**Table 3 materials-12-01046-t003:** Performance index summary of the expansion agent.

Type	Magnesium Oxide Content (%)	Total Alkali Content (%)	Chloride Ion (%)	Surface Areas (m^2^/kg)	Initial Setting (min)	Final Setting (min)	Restrained Expansion Rate		Compressive Strength (MPa)	
							Water 7 d	Air 21 d	7 d	28 d
UEA-H	3.38	0.58	0.009	316	174	259	0.036	−0.019	22.2	40.3

**Table 4 materials-12-01046-t004:** Performance index summary for the AN4000 polycarboxylic water-reducer and the AN1 air entraining agent.

Type	Water Reducing Ratio (%)	Bleeding Rate (%)	Air Content (%)	Air Content Changing within 1 h (%)	Shrinkage Ratio (%)	Initial Setting (min)	Final Setting (min)	Compressive Strength Ratio (%)		
								3 d	7 d	28 d
AN1	7.8	50	6.0	+1.3	121	+25	+30	96	97	95
AN4000	30.0	6.0	2.8	35	100	+105	+110	173	160	155

**Table 5 materials-12-01046-t005:** Mechanical properties of steel tubes.

Type	Tensile Strength (MPa)	Yield Strength (MPa)	Elongation (Min)	Shear Strength (MPa)	Elasticity Modulus (MPa)	Shear Modulus (MPa)	Coefficient of Linear Expansion (°C)	Mass Density (kg/m^3^)
Q345	560	347	26	182	2.04 × 10^5^	7.9 × 10^4^	1.2×10^−5^	7853

**Table 6 materials-12-01046-t006:** Summary of the experimental design on 12 concrete-filled steel tube (CFST) specimens.

Specimen	Lateral Restraint	Expansion Agent Dosage %	Steel Ratio %	Structure Size	Mixture Ratio Kg/m^3^						
				*(**D* × *t* × *H*) mm	Cement	Expansion Agent	Fine Aggregate	Coarse Aggregates	Fly Ash	Mineral Powder	Water
C-1	YES	4	3.8	φ140.0 × 1.3 × 350	368	19.6	740	1023	73	49	158.76
C-2	YES	8	3.8	φ140.0 × 1.3 × 350	368	39.2	740	1023	73	49	158.76
C-3	YES	12	3.8	φ140.0 × 1.3 × 350	368	58.8	740	1023	73	49	152.88
C-4	YES	4	6.6	φ140.0 × 2.2 × 350	368	19.6	740	1023	73	49	158.76
C-5	YES	8	6.6	φ140.0 × 2.2 × 350	368	39.2	740	1023	73	49	164.64
C-6	YES	12	6.6	φ140.0 × 2.2 × 350	368	58.8	740	1023	73	49	152.88
C-7	YES	4	9.2	φ140.0 × 3.0 × 350	368	19.6	740	1023	73	49	158.76
C-8	YES	8	9.2	φ140.0 × 3.0 × 350	368	39.2	740	1023	73	49	164.64
C-9	YES	12	9.2	φ140.0 × 3.0 × 350	368	58.8	740	1023	73	49	158.76
C-10	NO	4	0.0	φ135.6 × 0.0 × 350	368	19.6	740	1023	73	49	158.76
C-11	NO	8	0.0	φ135.6 × 0.0 × 350	368	19.6	740	1023	73	49	158.76
C-12	NO	12	0.0	φ135.6 × 0.0 × 350	368	19.6	740	1023	73	49	158.76

Note: *D* is the diameter of test structure; *t* is the thickness of the test structure; *H* is the height of the test specimen.

**Table 7 materials-12-01046-t007:** The effects of lateral restraint and different steel ratios on the creep of concrete.

Time (h)	C-1/C-10	C-4/C-10	C-7/C-10	C-2/C-11	C-5/C-11	C-8/C-11	C-3/C-12	C-6/C-12	C-9/C-12
240	0.19	0.08	0.11	0.28	0.13	0.12	0.28	0.14	0.15
1200	0.23	0.12	0.11	0.25	0.16	0.12	0.26	0.17	0.12
2400	0.23	0.14	0.10	0.23	0.17	0.11	0.24	0.18	0.12
3600	0.21	0.15	0.10	0.21	0.15	0.10	0.22	0.17	0.11
4800	0.19	0.13	0.09	0.20	0.14	0.09	0.20	0.15	0.10
7200	0.18	0.13	0.08	0.19	0.14	0.09	0.19	0.14	0.09
9600	0.18	0.13	0.08	0.18	0.14	0.09	0.18	0.14	0.09
11520	0.18	0.13	0.08	0.18	0.13	0.09	0.18	0.14	0.09

**Table 8 materials-12-01046-t008:** Summary of parameters used for studying the confinement effect of the combined structure.

Specimen	Steel Ratio %	Standard Confinement Coefficient ξ	Standard Value of Combined Compressive Strength (MPa)	Distribution Coefficient of Concrete Load	Distribution Coefficient of Steel Tube Load	Stress Ratio
C-2	3.8	0.253	53.002	0.823	0.177	0.288
C-5	6.6	0.437	59.810	0.729	0.271	0.256
C-8	9.2	0.606	65.767	0.660	0.340	0.232
C-10	0.0	0.000	43.026	1.000	0.000	0.379

**Table 9 materials-12-01046-t009:** Different effects of expansion agent concentration on the creep of concrete.

Time/h	C-2/C-1	C-3/C-1	C-5/C-4	C-6/C-4	C-8/C-7	C-9/C-7	C-11/C-10	C-12/C-10
240	1.267	1.158	1.419	1.419	0.976	1.107	0.841	0.778
1200	0.910	0.863	1.128	1.090	0.933	0.891	0.848	0.788
2400	0.876	0.831	1.012	0.964	0.942	0.900	0.879	0.787
3600	0.867	0.831	0.932	0.905	0.954	0.875	0.886	0.794
4800	0.880	0.831	0.915	0.880	0.942	0.900	0.855	0.789
7200	0.865	0.831	0.895	0.869	0.932	0.915	0.860	0.805
9600	0.876	0.825	0.915	0.883	0.934	0.893	0.861	0.814
11520	0.876	0.831	0.900	0.874	0.942	0.900	0.864	0.817

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
