# Peer review of "Experimental Studies on the Effect of Properties and Micro-Structure on the Creep of Concrete-Filled Steel Tubes"

_materials, 2019, doi:10.3390/ma12071046_

Round 1

Reviewer 1 Report

This paper presents an experimental study on the influence of lateral steel confinement and expansive agent (lateral prestress) on concrete creep for concrete filled steel tube specimens. The manuscript is generally well configured with adequate figures and tables to explain the experimental program and results. However, the text is noticeably weak and in some areas is simply causing confusion to the readers. This reviewer recommends the authors perform significant improvements to the text, the additional notes below will help address this:

Examples of text that requires attention is given below. Please be aware these are only a few examples and many further areas of improvement are required for this paper to be considered acceptable.

1-      Clumsy text exists that does not accurately describe your experiments. For example, Line 18 in the Abstract has “lateral restraint improves the strength of concrete tubes”. This is fundamentally wrong as there are no concrete tubes in the experimental program.

2-      Throughout the introduction many vague points are mentioned without detailed explanation. An example of this is Lines 66-68 which states “… which is different from creep in ordinary concrete.” A statement like this is not useful to the readers without a clear explanation of why or how it is different.

3-      Similar to point 2, Line 73 includes discussion of a creep model for concrete pipes, however this seems too disconnected from your selected research area of concrete creep for CFSTs.

4-      Table 6 has a title which states 12 specimens, although only 10 are presented.

5-      The conclusions are not strong. At line 304 the statement “Meanwhile, creep ratio of the specimens with lateral restraints decreases with steel rate growth” is simply confusing. Additionally, Line 312 refers to “content of expansion” which is not clear if the authors are referring to amount of expansion agent, or type of content used in the expansion agent. Finally, Line 315 has “CSFT” instead of “CFST”, which indicates further lack of attention to detail.

Author Response

Dear Editors and Reviewers:

Thank you for your letter and reviewers’ comments concerning about our manuscript entitled “Experimental studies on the effect of properties and micro-structure on the creep of concrete-filled steel tubes” (ID:materials-463880). As can be seen from the review comments, the reviewers had put forward a lot of valuable opinions and advice throughout this article. It’s quite beneficial for me to see these comments and I really appreciate for that. Those comments are all valuable and very helpful for revising and improving our paper. We have gone through those comments carefully and made corresponding corrections on the manuscript to address the reviewers’ comments and meet the requirements with approval. Revised parts are marked in red with “Tack Changes” function in the manuscript. The main corrections made in the manuscript and responds to the reviewer’s comments are listed below:

Responds to the reviewer’s comments:

Point 1: Clumsy text exists that does not accurately describe your experiments. For example, Line 18 in the Abstract has “lateral restraint improves the strength of concrete tubes”. This is fundamentally wrong as there are no concrete tubes in the experimental program.

Response 1: Sorry for the misusage of “concrete tubes”. The correct description should be: lateral restraint improves the strength of concrete.

Point 2: Throughout the introduction many vague points are mentioned without detailed explanation. An example of this is Lines 66-68 which states “… which is different from creep in ordinary concrete.” A statement like this is not useful to the readers without a clear explanation of why or how it is different.

Response 2:This statement, “… which is different from creep in ordinary concrete”, is made from the experimental research regarding creep of concrete filled steel tube is different from ordinary concrete. The author has discussed this statement in the original submission, now at Lines 106-107 in the revised manuscript, which states ‘The effects of expansion agents and the lateral restraint effect by steel tubes on creep are different from those of ordinary concrete’. Therefore, the statement “… which is different from creep in ordinary concrete” which is at lines 66-68 in the original submission was deleted.

This research has mainly investigated the creep of concrete with two lateral restraint conditions (with and without lateral restraints) and four steel ratios (0.0%, 3.8%, 6.6% and 9.2%) to study the difference in creep characteristics of CFST and ordinary concrete. Detailed experimental results analysis and discussion were made to state why it’s different in Section 3.1 of the manuscript. The major conclusions on how the creep characteristics of CFST are different from that of the ordinary concrete were listed in Section 5.

Point 3: Similar to point 2, Line 73 includes discussion of a creep model for concrete pipes, however this seems too disconnected from your selected research area of concrete creep for CFSTs.

Response 3:To provide sufficient information and background with relevant references to the introduction, we have re-written this part according to the reviewer’s suggestion. We have deleted some irrelevant contents which include Line 73: “Wang [19] develops a creep model of concrete pipe for axial compression members made of high-performance concrete” and added the most relevant and the latest international research results. Revised parts are marked in red in the introduction section.

Point 4: Table 6 has a title which states 12 specimens, although only 10 are presented.

Response 4:It has been corrected to 10 specimens. Sorry for the typo.

Point 5: The conclusions are not strong. At line 304 the statement “Meanwhile, creep ratio of the specimens with lateral restraints decreases with steel rate growth” is simply confusing. Additionally, Line 312 refers to “content of expansion” which is not clear if the authors are referring to amount of expansion agent, or type of content used in the expansion agent. Finally, Line 315 has “CSFT” instead of “CFST”, which indicates further lack of attention to detail.

Response 5:To make the conclusions to be clearer and stronger, considering reviewer’s suggestions, we have re-written this part as “Meanwhile, the creep ratio of CFST specimens decreases as the steel ratio increases.” The major function of steel tube is to provide lateral restraints for the filled concrete and so the authors use the phrase “CFST specimens” instead of “the specimens with lateral restraints” to clarify the meaning of this conclusion and make it to be simpler.

As the reviewer suggested, the authors change “content of expansion” to “expansion agent concentrations” to clarify the meaning. In this paper, it is referring to amount of expansion agent, not the type of content used in the expansion agent.

We are very sorry for the typo ‘CSFT, and it has been revised to “CFST” instead of “CSFT”. To pay further attention to the detail, we have carefully and thoroughly checked the whole manuscript. Revised parts are marked in red in the introduction.

We tried our best to improve the manuscript and made some changes with “Tack Changes” in the manuscript. These changes have no influence on the major content and framework of the paper. And we did not list all the changes in this letter, but they can be found in red in the revised manuscript.

We really appreciate for the editors and reviewers’ work and valuable suggestions on improving this research article, and we hope the corrections that we made could meet the requirements with approval.

Once again, thanks a lot for all the comments and suggestions.

Many thanks with the best wishes.

        Rongling Zhang

Reviewer 2 Report

This paper would need substantial changes and improvement before publication. The English is too poor and often cannot be understood. The stress distribution between concrete and steel tube should be taken into consideration. What is the meaning of Fig. 4 for instance? The list of references is filled with Chinese authors, but most relevant international authors are not taken into consideration.

Author Response

Dear Editors and Reviewers:

Thank you for your letter and reviewers’ comments concerning about our manuscript entitled “Experimental studies on the effect of properties and micro-structure on the creep of concrete-filled steel tubes” (ID:materials-463880). As can be seen from the review comments, the reviewers had put forward a lot of valuable opinions and advice throughout this article. It’s quite beneficial for me to see these comments and I really appreciate for that. Those comments are all valuable and very helpful for revising and improving our paper. We have gone through those comments carefully and made corresponding corrections on the manuscript to address the reviewers’ comments and meet the requirements with approval. Revised parts are marked in red with “Tack Changes” function in the manuscript. The main corrections made in the manuscript and responds to the reviewer’s comments are listed below:

Responds to the reviewer’s comments:

Point 1: This paper would need substantial changes and improvement before publication. The English is too poor and often cannot be understood.

Response 1: We are very sorry for the confused content and poor English. Because English is not my mother language, but I seriously and thoroughly checked and modified the whole article. I have asked another two coauthors of this paper, Dr. Li Jia, who got her PhD in the United States, and Yu Wang, a PhD student from the UK, to proofread and improve the paper. Alternatively, MDPI provides an English editing service checking grammar, spelling, punctuation and some improvement of style for this paper. We tried our best to improve the manuscript including the English writing. Revised parts are marked in red in the paper.

Point 2: The stress distribution between concrete and steel tube should be taken into consideration.

Response 2: As the reviewer suggested, we completely agree that the stress distribution between concrete and steel tube is a very important point and should be taken into consideration in more detailed study on the CFST. We have performed further analysis and calculations on the stress distribution between concrete and steel tube.

In this paper, based on the ferrule effect of the combined structure, the mechanical properties of the concrete under different lateral restraints are calculated based on the confinement effect of the combined structure. The influence of the steel tube’s lateral restraints on the creep characteristics of the concrete was studied via detailed calculation and analysis considering the load distribution between concrete and steel tube. The results of the analysis are summarized in Table 8. The standard confinement coefficient and the standard value of combined compressive strength increased with increasing steel ratio. The stress ratio of the CFST specimen decreased as the steel ratio increased. The distribution coefficient of concrete load increased and that of the steel tube load decreased with increasing steel ratio. That is, the difference in lateral restraints caused the difference in stress distribution between the concrete and steel tubes. This also shows why lateral restraint influences the creep characteristics of concrete.

Table 8. Summary of Parameters Used for Studying the Confinement Effect of the Combined Structure

Specimen

Steel  

Ratio   %

Standard  

Confinement  

Coefficient
  ξ

Standard   Value of

Combined   Compressive

 Strength (MPa)

Distribution  

Coefficient   of

Concrete   Load

Distribution  

Coefficient   of

Steel   Tube Load

Stress   Ratio

C-2

3.8

0.253

53.002

0.823

0.177

0.288

C-5

6.6

0.437

59.810

0.729

0.271

0.256

C-8

9.2

0.606

65.767

0.660

0.340

0.232

C-10

0.0

0.000

43.026

1.000

0.000

0.379

Point 3: What is the meaning of Fig. 4 for instance?

Response 3: The authors have described the experiment at Lines 285-287 in the revised manuscript. We are very sorry for the unclear description on the meaning of Figure 4. We changed the position of Lines 285-287 in the revised manuscript and added relevant description to clarify the experimental process.

The added description is “Plastic wraps and tapes were used to seal-cure the cube specimens. The top two layers of the cube specimens were ordinary concrete specimens, and the bottom layer consisted of four concrete specimens where steel molds were not removed. These steel molds were used to simulate the concrete with lateral restraints. The weight of the top two layers also restrained the vertical expansion of the specimens at the bottom layer. The seal curing setup of the specimens with different lateral restraint conditions is shown in Figure 4.”  

Point 4: The list of references is filled with Chinese authors, but most relevant international authors are not taken into consideration.

Response 4: After double checked the manuscript and the corresponding references, the authors searched and carefully studied more relevant and more recent research work of the international authors on the relevant topic. Finally, the authors added the most relevant international research results into the introduction section and citation of those work were listed in the references section and marked in red.

We tried our best to improve the manuscript and made some changes with “Tack Changes” in the manuscript. These changes have no influence on the major content and framework of the paper. And we did not list all the changes in this letter, but they can be found in red in the revised manuscript.

We really appreciate for the editors and reviewers’ work and valuable suggestions on improving this research article, and we hope the corrections that we made could meet the requirements with approval.

Once again, thanks a lot for all the comments and suggestions.

Many thanks with the best wishes.

       Rongling Zhang

Round 2

Reviewer 1 Report

Noticeable improvements are evident for this version of the manuscript. Most noticeably is the level of English. This reviewer acknowledges the provided "track changes" which documents significant changes. This has reduced the general confusion present in the previous version and improves the overall paper’s strength. A few minor oversights exist which are listed below:

1.       The previous comment (Point 1) in regards to clumsy text still applies. The revised text is still fundamentally incorrect. The concrete strength does not change due to the presence of lateral restraint. The strength of the system (concrete + lateral restraint) improves to become greater than the strength of the concrete. I.e. the system still resists further load after the concrete ultimate strength is surpassed. Adjust the wording here to avoid confusing the reader.

2.      Line 179 states “The results of all 10 creep tests ..” Should this be 12 or 10 here? Correct this if required.

Author Response

Dear Editors and Reviewers:

Thank you for your letter and reviewers’ comments concerning about our manuscript entitled “Experimental studies on the effect of properties and micro-structure on the creep of concrete-filled steel tubes” (ID:materials-463880). I really appreciate for the reviewer had put forward valuable opinions. That comment is really valuable for us to further improve our paper. We have made corresponding corrections on the manuscript to address the reviewer’s comment and meet the requirements with approval. Revised parts are marked in red with “Tack Changes” function in the manuscript.

Responds to the reviewer’s comments:

Point 1:  The previous comment (Point 1) in regards to clumsy text still applies. The revised text is still fundamentally incorrect. The concrete strength does not change due to the presence of lateral restraint. The strength of the system (concrete + lateral restraint) improves to become greater than the strength of the concrete. I.e. the system still resists further load after the concrete ultimate strength is surpassed. Adjust the wording here to avoid confusing the reader.

Response 1: According to the valuable suggestions from the reviewer, we clarified the findings from the test results and updated the corresponding description to “The test results show that the lateral restraint improves the strength of the system (concrete-filled steel tubes) which resists further load after the concrete ultimate strength is surpassed and reduces the creep.” to avoid confusing the reader.

Point 2:  Line 179 states “The results of all 10 creep tests.” Should this be 12 or 10 here? Correct this if required.

Response 2: Thank you for pointing out this typo. We updated the tile and content of Table 6 at Line 184-186 to complete the information for all 12 specimens’ properties, and we corrected the description at Line 179 in the last updated manuscript to “The results of all 12 creep tests.” now at Line 194.

We tried our best to improve the manuscript and made some changes with “Tack Changes” in the manuscript. These changes have no influence on the major content and framework of the paper.

We really appreciate for the editors and reviewers’ work and valuable suggestions on further improving this research article, and we hope the corrections that we made could meet the requirements with approval.

Once again, thanks a lot for all the comments and suggestions.

Many thanks with the best wishes.

       Rongling Zhang

Reviewer 2 Report

The paper was substantially improved. There are still some weak points as for instance Fig. 4, The meaning of this figure can not be understood.

Author Response

Dear Editors and Reviewers:

Thank you for your letter and reviewers’ comments concerning about our manuscript entitled “Experimental studies on the effect of properties and micro-structure on the creep of concrete-filled steel tubes” (ID:materials-463880). I really appreciate for the reviewer had put forward valuable opinions. That comment is really valuable for us to further improve our paper. We have made corresponding corrections on the manuscript to address the reviewer’s comment and meet the requirements with approval. Revised parts are marked in red with “Tack Changes” function in the manuscript.

Responds to the reviewer’s comments:

Point 1: The paper was substantially improved. There are still some weak points as for instance Fig. 4, The meaning of this figure can not be understood.

Response 1: To clearly describe the test process and to clear explanation of what is the meaning of Figure 4. We moved Figure 4 to part 4 “The Analysis of the Micro Pore Structure”, now is Figure 6, and rewrote the description at Lines 285-291 for further clarify the meaning of the figure.

The updated description is “Figure 6 shows that the cube specimens were curing for pore structure analysis. Plastic film was used to seal all the cube specimens to simulate sealing effect of steel tubes on the core concrete of the CFST specimens. Concrete specimens at the bottom layer were not demoulded to provide the lateral restraints to simulate the effect of steel tube on the core concrete. Concrete specimens at the top two layers were demouled to simulate the ordinary concrete without lateral restraints. Moreover, the top two layers provided vertical restraint for the bottom layer. Finally, the pore structures of the concrete specimens were analyzed at the age of 28 days.” 

We tried our best to improve the manuscript and made some changes with “Tack Changes” in the manuscript. These changes have no influence on the major content and framework of the paper.

We really appreciate for the editors and reviewers’ work and valuable suggestions on further improving this research article, and we hope the corrections that we made could meet the requirements with approval.

Once again, thanks a lot for all the comments and suggestions.

Many thanks with the best wishes.

        Rongling Zhang
